# Validation of a Classification Algorithm for Chronic Kidney Disease Based on Health Information Systems

**DOI:** 10.3390/jcm11102711

**Published:** 2022-05-11

**Authors:** Pietro Manuel Ferraro, Nera Agabiti, Laura Angelici, Silvia Cascini, Anna Maria Bargagli, Marina Davoli, Giovanni Gambaro, Claudia Marino

**Affiliations:** 1U.O.S. Terapia Conservativa della Malattia Renale Cronica, Fondazione Policlinico Universitario A. Gemelli IRCCS, 00168 Roma, Italy; pietromanuel.ferraro@unicatt.it; 2Università Cattolica del Sacro Cuore, 00168 Roma, Italy; 3Department of Epidemiology Regional Health Service—Lazio, 00147 Rome, Italy; l.angelici@deplazio.it (L.A.); s.cascini@deplazio.it (S.C.); a.bargagli@deplazio.it (A.M.B.); m.davoli@deplazio.it (M.D.); c.marino@deplazio.it (C.M.); 4Renal Unit, Division of Nephrology and Dialysis, Department of Medicine, University of Verona, 37126 Verona, Italy; giovanni.gambaro@hotmail.it

**Keywords:** chronic kidney disease, administrative data, serum creatinine measurements, validation, algorithm, eGFR

## Abstract

Background: Chronic kidney disease (CKD) is a common condition, characterized by high burden of comorbidities, mortality and costs. There is a need for developing and validating algorithm for the diagnosis of CKD based on administrative data. Methods: We validated our previously developed algorithm that used administrative data of the Lazio Region (central Italy) to define the presence of CKD on the basis of serum creatinine measurements performed between 2012 and 2015 at the Policlinico Gemelli Hospital. CKD and advanced CKD were defined according to eGFR (<60 and <30 mL/min/1.73 m^2^, respectively). Sensitivity, specificity, positive and negative predictive values (PPV/NPV) were computed. Results: During the time span of the study, 30,493 adult participants residing in the Lazio Region had undergone at least 2 serum creatinine measurements separated by at least 3 months. CKD and advanced CKD were present in 11.1% and 2.0% of the study population, respectively. The performance of the algorithm in the identification of CKD was high, with a sensitivity of 51.0%, specificity of 96.5%, PPV of 64.5% and NPV of 94.0%. Using advanced CKD, sensitivity was 62.9% (95% CI 59.0, 66.8), specificity 98.1%, PPV 40.4% and NPV 99.3%. Conclusion: The algorithm based on administrative data has high specificity and adequate performance for more advanced CKD; it can be used to obtain estimates of prevalence of CKD and to perform epidemiological research.

## 1. Introduction

Chronic kidney disease (CKD) is a common condition [1,2] characterized by the progressive decline of the physiological functions of the kidney and a number of abnormalities including hormonal derangements, accumulation of waste products and water and electrolyte disturbances [3]. It is a known risk factor for the development of serious adverse clinical outcomes, including cardiovascular morbidity and mortality and need for kidney replacement therapy. Hence, it is not surprising that CKD is also characterized by high costs for the health systems [4].

For the correct allocation of resources and implementation of prevention and management strategies, national and regional government bodies need to rely on estimates of disease burden. For CKD, precise estimation of prevalence is made difficult by the silent course characterizing the initial stages of the disease, before the development of complications such as cardiovascular events, anemia, fractures, metabolic acidosis, electrolyte disorders, symptoms of intoxication that finally force the patient to seek medical attention. The current definition of CKD relies on biochemical or imaging evidence of impaired kidney structure or function lasting for at least 3 months [5]. In the absence of other abnormalities, a persistently reduced estimated glomerular filtration rate (eGFR) < 60 mL/min/1.73 m^2^ allows the diagnosis of CKD. It is also possible to stage disease severity according to values of eGFR and albumin-to-creatinine ratio (Appendix A). To obtain estimates of prevalence based on administrative data, thus overcoming the limitations of having to rely on repeated determinations of eGFR (whose estimation is in turn based on serum creatinine), we previously developed and published an algorithm based on health information systems of the Lazio Region, Italy [6]. Such approach has several advantages, including the possibility to export the algorithm to health systems with similar information systems and the immediate availability of data. The purpose of this study was to determine the validity of our algorithm based on administrative data for identifying patients with CKD compared to the reference standard of eGFR.

## 2. Materials and Methods

### 2.1. Data Sources

#### 2.1.1. Laboratory Data

Information on serum creatinine was obtained from the central laboratory of the Gemelli Hospital, an academic medical centre in Rome, Lazio, as part of a study on clinical outcomes of patients affected with CKD (local Ethics Committe protocol number 47284/17). The Hospital is a tertiary care centre and serves patients from southern and central Italy. The data contain all laboratory measurements prescribed during hospitalization or emergency room access or ambulatory care. Those data were used to identify participants with CKD.

#### 2.1.2. Health Administrative Data

The procedure followed to generate the algorithm has been already described [6]. Briefly, Lazio Regional Health Information Systems (HIS) used to perform the CKD algorithms were the following: hospital discharge registry, ticket exemption registry (a registry of all residents who are entitled to co-pay fee exemption for particular conditions, e.g., disability, chronic diseases, low income or old age), outpatient specialist service information system and drug dispensing registry. Regional health assistance file was used to assess the residence of individuals and regional mortality registry was used to assess vital status.

### 2.2. Study Population

All individuals who had undergone at least 2 serum creatinine measurements separated by at least 3 months in the period 1 January 2012–31 January 2015 were potential subjects to be included. For each participant, the last creatinine measurement was selected, and the date of the measurement was defined as index date. Participants younger than 19 years and those who were not resident in the Lazio Region during the 5 years before and 1 year after the index date were excluded.

### 2.3. Measure of Kidney Function and Definition of CKD

eGFR was estimated using the recently developed race-free CKD-EPI equation [7]. CKD and advanced CKD were defined as the presence of the most recent eGFR values < 60 and < 30 mL/min/1.73 m^2^, respectively, separated by at least 3 months from another eGFR value < 60 mL/min/1.73 m^2^. They were used as the working standard indicating presence of CKD or presence of advanced CKD. Serum creatinine was measured with the enzymatic method; measurements from both inpatient and outpatient events were used.

### 2.4. Health Administrative Data Definitions of CKD

Participants were linked by the regional anonymous personal code with the HIS necessary to implemented our previously developed algorithm for the diagnosis of CKD in a period of 6 years around the index date (5 year before and 1 year after). In Appendix A were presented all codes used to perform the algorithm, for each HIS [6].

Furthermore, among patients with CKD, those with advanced stages of CKD were identified. Higher severity patients were subjects who during the selection period had undergone chronic dialysis or at least one kidney transplant or one hospitalization with diagnosis code of CKD stage G4 or greater, or who had been prescribed at least one of the drugs selected in the algorithm (erythropoietin, darbepoetin alfa, methoxy polyethylene glycol-epoetin beta, polystyrene sulfonate, sevelamer, lanthanum carbonate, sucroferric oxyhydroxide) [6].

### 2.5. Statistical Analysis

Descriptive analysis was used to describe demographic characteristics of the study population, stratified by CKD stages assessed at the last serum creatinine measurement. Furthermore, the study population was described according to the presence or absence of CKD-working standard and CKD-algorithm, both for CKD cases and advanced cases. Estimates of sensitivity, specificity, positive predictive value (PPV) and negative predictive value (NPV) with 95% confidence intervals (CIs) were generated using 2 × 2 tables. All validity measurements were calculated by sex and age groups (19–44, 45–64, 65–74, 75–84, 85+ years).

A sensitivity analysis was performed restricting the study population to those with serum creatinine measurements in the outpatient setting.

Statistical analyses were done using SAS version 9.2 (SAS Institute Inc., Cary, NC, USA).

## 3. Results

During the time span between 1 January 2012 and 31 January 2015, data from 800,203 serum creatinine measurements from 198,179 individuals were performed. Of those, 45,436 participants had undergone at least 2 serum creatinine measurements separated by at least 3 months. At index date, 30,493 individuals were older than 18 years and resident in the Lazio Region during the period 5 years before and 1 year after the index date (Figure 1). Men were 40.8% and mean age was 57.3 years (SD 17.9), with men being on average older than women (60.2 vs. 55.4 years, *p*-value *t* test < 0.0001). Their baseline characteristics are presented in Table 1.

Based on the definition of CKD as eGFR consistently < 60 mL/min/1.73 m^2^, CKD was present in 11.1% (95% CI 10.8, 11.5) of the study population. The prevalence was higher among men compared with women (13.6% vs. 9.4%, *p*-value χ^2^ < 0.0001) and increased with age, ranging from 1.1% (95% CI 0.9, 1.3) among those aged between 19 and 44 years up to 47.6% (95% CI 44.9, 50.3) among those aged 85+ years. Advanced CKD (eGFR < 30 mL/min/1.73 m^2^) was present in 2.0% (95% CI 1.8, 2.1) of the study population, with trends similar to CKD across sex and age (Table 2). The prevalence of CKD and severe CKD based on the diagnostic algorithm was 8.8% (95% CI 8.5, 9.1) and 3.1% (CI 95% 2.9, 3.3), respectively, with trends similar to CKD patients across sex and age class (Table 2).

The performance of the algorithm in the identification of CKD was high, with a sensitivity of 51.0% (95% CI 49.3, 52.6), specificity of 96.5% (95% CI 96.3, 96.7), PPV of 64.5% (95% CI 62.9, 66.1) and NPV of 94.0% (95% CI 93.8, 94.2). Sensitivity was higher among men than women (60.0% vs. 41.9%) while specificity was slightly higher among women (97.9 vs. 94.6) (Table 3); both validity measures decreased with increase of age classes: sensitivity was 86.0% in age group 19–44 years and 46.3% in age group 85+ years and specificity was 98.1 and 93.6 in the same age classes (Figure 2). Using advanced CKD, all validity parameters improved except for a slight reduction in PPV. Sensitivity was 62.9% (95% CI 59.0, 66.8), specificity 98.1% (95% CI 98.0, 98.3), PPV 40.4% (95% CI 37.9, 42.9) and NPV 99.3% (95% CI 99.2, 99.3) (Table 3). The differences between sex and age groups were maintained (Figure 2).

Results were substantially unmodified after restricting the analysis to serum creatinine determinations performed in the outpatient setting (Appendix A).

## 4. Discussion

In our study, we sought to validate our previously developed algorithm based on administrative data. After linkage with a large database of inpatient and outpatient serum creatinine measurements, we defined the presence of CKD according to repeated eGFR determinations and computed the proportion of participants who were correctly classified by the algorithm as affected or not affected by CKD during a time span covering of 5 years before and 1 year after the date of CKD status definition.

Overall, we found a prevalence of CKD of about 11%. This figure is at odds with previous reports of CKD prevalence in the Italian population. For instance, the prevalence of CKD in the Northeastern Italy INCIPE study was about 13% overall, but only 6.7% when considering stages G3 or worse, which is the definition of CKD used in our study [1]. This discrepancy might in part be due to differences in the study populations, with the INCIPE cohort including a random sample of the general population whereas our sample was selected based on having at least one laboratory measurement performed at the Gemelli Hospital: this approach might have selected a less healthy population in our study. Another potential explanation is that a single determination of serum creatinine might have misclassified some INCIPE participants as non-CKD. Similar considerations apply to the CARHES study, a national survey reporting an even lower prevalence of CKD stage G3 or worse of about 2.9% [8].

We found that 1724 (51.0%) of the 3384 CKD participants and 26,160 (96.5%) of the 27,109 non-CKD participants were correctly classified by the algorithm. When focusing on advanced CKD, 378 (62.9%) of the 601 CKD participants and 29,334 (98.1%) of the 29,892 non-CKD participants were correctly classified. Taken together, these findings suggest that the overall performance of the algorithm is adequate, although it tends to underestimate true CKD, especially the milder forms. This is not unexpected given that the algorithm is based on utilization of healthcare resources, and initial and milder stages of CKD, being generally asymptomatic or paucisymptomatic, might remain undetected. Depending on the setting and on whether a broader or stricter definition of CKD is desired, the algorithm for the prediction of more severe CKD could be used, resulting in an increase in both sensitivity and specificity. However, the algorithm appears to be cost-effective given that no additional resources (both in monetary and person-time terms) are needed to generate its output, the input being constituted of mandatory, administrative data available to all regional health systems in Italy. A further advantage of the algorithm is the possibility to provide estimates of kidney function status for epidemiological research studies, with the possibility to use such estimates as exposure/confounder/outcome variables.

To date, several validation studies have been published, mainly in North America; however, due to heterogeneity of methodology and a variety of data sources, comparing results is difficult. In the systematic review of Vlasschaert et al., the authors conducted a comprehensive global review of 25 studies, quantifying the accuracy of codes for acute kidney injury and CKD. Given the heterogeneity of results, they did not perform a meta-analysis and concluded that the administrative database analyses have utility, but must be conducted and interpreted judiciously to avoid bias arising from poor code validity [9]. In another review, Grams at al. explain that their results show large variability in the accuracy of data items, depending on the variable of interest, study population and the comparison gold standard; so additional research is required to investigate sources of this variation and conclude that existing data sources need careful scrutiny before use in any research effort and that an electronic medical record integrated with laboratory and vital status data would greatly facilitate clinical and epidemiologic research [10]. Moreover, an Italian review (published in 2019) concludes on the need for further effort to improve algorithms for identification of CKD especially for early stages [11]. We found a PPV for eGFR < 60 mL/min/1.73 m^2^ lower than that reported in a recent large study from the US (64.5% vs. 84.4%), but trends across age classes and gender were similar [12]. In a study conducted in Canada (2020) in a population of just over 400 individuals, different algorithms were validated to identify patients with eGFR < 30 mL/min/1.73 m^2^, not on kidney replacement treatment (dialysis or kidney transplantation), authors found sensitivity and PPV higher and specificity and VPN lower than in our study [13].

We found that the performance of our algorithm was affected by sex: in particular, sensitivity was higher among men and specificity among women. This phenomenon could be due to differential access to healthcare; for instance, previous work from our group demonstrated that women have significantly lower odds of arteriovenous fistula placement compared with men [14]. Alternatively, women might develop CKD symptoms and signs at lower GFR values compared with men; this is a field in which active research is warranted [15,16].

We also found that age affected the performance of our algorithm, with both sensitivity and specificity reduced among older participants; this could be due, again, to differential access to healthcare across age groups, and/or to potential overestimation of CKD among older participants, giving rise to a higher proportion of CKD cases that would not need specific access to kidney care. However, alternative explanations could be hypothesized; for instance, older individuals might already have ticket exemptions for other conditions, or not receive a discharge diagnosis of CKD due to the number and severity of other comorbidities (thus diminishing the likelihood of being flagged with those criteria). The topic of CKD definition among the elderly is debated, and our results strengthen the idea that age-specific cutoffs for the definition of CKD should be used [17].

Our study has several strengths, including the large sample size accrued over a long time span, the possibility to analyze separately outpatient serum creatinine measurements and CKD defined at different thresholds, the implementation of recently developed eGFR equations and of the working standard definition of CKD, requiring chronic GFR reduction over time; with regard to the latter point, previous studies have shown that using a single serum creatinine measurement might have a significant impact on the definition of CKD [18]. Our study also has limitations, including the non-random sampling (individuals who elect to access the laboratory of a tertiary hospital might not represent the general adult population of the Lazio Region), lack of information on race and socio-economic status, and relatively low sensitivity especially for moderate CKD, causing a non-trivial rate of false negatives. Administrative data are known to have limitations in retrieving details in clinical information however the impact could be limited because our algorithm is used for epidemiological research and not for a diagnostic test.

## 5. Conclusions

In conclusion, our previously developed algorithm based on administrative data has high specificity and adequate performance for more advanced CKD and can be used to obtain estimates of prevalence of CKD and to perform epidemiological research. Future studies will include the implementation of more detailed laboratory data such as serum cystatin C and urinary albumin excretion as well as the extension of our model to a multi-regional setting to obtain and compare estimates of CKD at the national level.

## Figures and Tables

**Figure 1 jcm-11-02711-f001:**
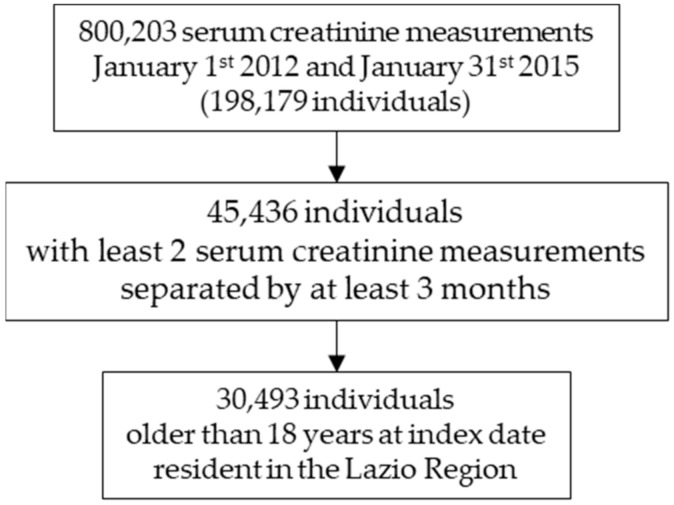
Flow-chart of study population selection.

**Figure 2 jcm-11-02711-f002:**
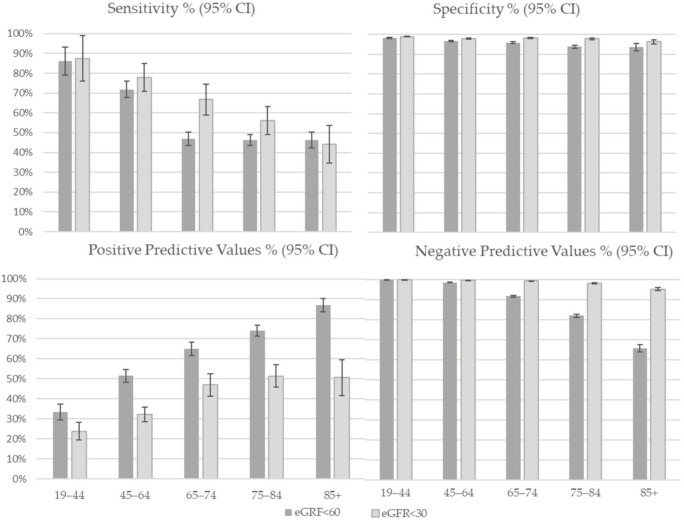
Validity of CKD definitions on administrative data compared with the eGFR reference standard, by age class.

**Table 1 jcm-11-02711-t001:** Distribution of Chronic Kidney Disease CKD stages based on the CKD-EPI equation *, by sex.

	Total		Male		Female	
	N	%	N	%	N	%
	30,493		12,427	40.8	18,066	59.2
**Age, years (mean (SD))**	57.3	(17.9)	60.2	(16.9)	55.4	(8.3)
**Age class (years)**						
**19–44**	8354	27.4	2400	19.3	5954	33.0
**45–64**	9826	32.2	4242	34.1	5584	30.9
**65–74**	6179	20.3	2977	24.0	3202	17.7
**75–84**	4827	15.8	2275	18.3	2552	14.1
**85+**	1307	4.3	533	4.3	774	4.3
**CKD stage**						
**eGFR ≥ 60 mL/min/1.73 m^2^**	27,109	88.9	10,737	86.4	16,372	90.6
**eGFR 45–59 mL/min/1.73 m^2^**	1749	5.7	859	6.9	890	4.9
**eGFR 30–44 mL/min/1.73 m^2^**	1034	3.4	524	4.2	510	2.8
**eGFR 29–15 mL/min/1.73 m^2^**	406	1.3	190	1.5	216	1.2
**eGFR < 15 mL/min/1.73 m^2^**	195	0.6	117	0.9	78	0.4

* CKD-EPI equation is the recently developed race-free equation to estimate the glomerular filtration rate (eGFR) [7].

**Table 2 jcm-11-02711-t002:** Characteristics of patients with or without Chronic Kidney Disease CKD-working standard (WS) and CKD-algorithm (AL) and prevalence of CKD in both definitions, by sex and age class.

	No CKD (WS)		CKD (WS)		No CKD (WS)		CKD (WS)		CKD (WS)			CKD (AL)		
	No CKD (AL)		No CKD (AL)		CKD (AL)		CKD (AL)		Prevalence %	95% CIs		Prevalence %	95% CIs	
	N	%	N	%	N	%	N	%		Inf	Sup		Inf	Sup
**eGFR * < 60 mL/min/1.73 m^2^/algorithm for identification of CKD**														
**Total**	**26,160**		**1660**		**949**		**1724**		**11.1**	**10.8**	**11.5**	**8.8**	**8.5**	**9.1**
**Sex**														
**Male**	10,159	38.8	676	40.7	578	60.9	1014	58.8	13.6	13.0	14.2	12.8	12.2	13.4
**Female**	16,001	61.2	984	59.3	371	39.1	710	41.2	9.4	9.0	9.8	6.0	5.6	6.3
**Age class (years)**														
**19–44**	8102	31.0	13	0.8	159	16.8	80	4.6	1.1	0.9	1.3	2.9	2.5	3.2
**45–64**	9060	34.6	129	7.8	309	32.6	328	19.0	4.7	4.2	5.1	6.5	6.0	7.0
**65–74**	5085	19.4	464	28.0	221	23.3	409	23.7	14.1	13.3	15.0	10.2	9.5	11.0
**75–84**	3272	12.5	720	43.4	216	22.8	619	35.9	27.7	26.5	29.0	17.3	16.2	18.4
**85+**	641	2.5	334	20.1	44	4.6	288	16.7	47.6	44.9	50.3	25.4	23.1	27.8
**eGFR * < 30 mL/min/1.73 m^2^/algorithm for identification of advanced CKD**														
**Total**	**29,334**		**223**		**558**		**378**		**2.0**	**1.8**	**2.1**	**3.1**	**2.9**	**3.3**
**Sex**														
**Male**	11,777	40.1	92	41.3	343	61.5	215	56.9	2.5	2.2	2.7	4.5	4.1	4.9
**Female**	17,557	59.9	131	58.7	215	38.5	163	43.1	1.6	1.4	1.8	2.1	1.9	2.3
**Age class (years)**														
**19–44**	8232	28.1	4	1.8	90	16.1	28	7.4	0.4	0.3	0.5	1.4	1.2	1.7
**45–64**	9480	32.3	29	13.0	215	38.5	102	27.0	1.3	1.1	1.6	3.2	2.9	3.6
**65–74**	5937	20.2	46	20.6	104	18.6	92	24.3	2.2	1.9	2.6	3.2	2.7	3.6
**75–84**	4527	15.4	86	38.6	104	18.6	110	29.1	4.1	3.5	4.6	4.4	3.9	5.0
**85+**	1158	3.9	58	26.0	45	8.1	46	12.2	8.0	6.5	9.4	7.0	5.6	8.3

* eGFR: estimate Glomerular Filtration Rate [7].

**Table 3 jcm-11-02711-t003:** Validity of Chronic Kidney Disease (CKD) definitions on administrative data compared with the estimate Glomerular Filtration Rate (eGFR) reference standard, by sex.

	Sensitivity	95% CIs		Specificity	95% CIs		PPV *	95% CIs		NPV ^	95% CIs	
**eGFR < 60 mL/min/1.73 m^2^/algorithm for identification of CKD**												
**Total**	**51.0**	**49.3**	**52.6**	**96.5**	**96.3**	**96.7**	**64.5**	**62.9**	**66.1**	**94.0**	**93.8**	**94.2**
**Sex**												
**Male**	60.0	57.7	62.3	94.6	94.2	95.0	63.7	61.7	65.7	93.8	93.4	94.1
**Female**	41.9	39.6	44.3	97.7	97.5	98.0	65.7	63.1	68.3	94.2	94.0	94.4
**eGFR < 30 mL/min/1.73 m^2^/algorithm for identification of advanced CKD**												
**Total**	**62.9**	**59.0**	**66.8**	**98.1**	**98.0**	**98.3**	**40.4**	**37.9**	**42.9**	**99.3**	**99.2**	**99.3**
**Sex**												
**Male**	70.0	64.9	75.2	97.2	96.9	97.5	38.5	35.5	41.6	99.2	99.1	99.4
**Female**	55.4	49.8	61.1	98.8	98.6	99.0	43.1	39.0	47.2	99.3	99.2	99.4

* PPV: positive predictive values. ^ NPV: negative predictive values.

## Data Availability

Not applicable.

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
