# Peer review of "Validation of a Classification Algorithm for Chronic Kidney Disease Based on Health Information Systems"

_jcm, 2022, doi:10.3390/jcm11102711_

Round 1

Reviewer 1 Report

The purpose of the manuscript is to validate their previously developed algorithm to define the presence of chronic kidney disease. A large number of adult patients were used in the experiment and acceptable levels of performance indices were found while detecting high CKD but not for mild CKD. 

Major points:

  1. In the abstract the authors claimed that: “adequate performance and can be used to obtain estimates of the prevalence of CKD”, a sensitivity of 51% can not be categorized as adequate. Authors need to rephrase their sentence about “adequate performance”.
  2. Authors claimed that “milder stages of CKD might remain undetected”, although the algorithm is cost-effective the authors need to explain what is the solution in case “milder CKD might remain undetected”.
  3. In the discussion, the authors discussed several validation studies but only a few of them were from the last 5 years. Authors need to do a more extensive literature survey and include all significant similar validation studies which happened in the last five years. Need to mention the strength and shortcomings of those studies as well.
  4. What are probable solutions for the limitations of this study which were mentioned by the authors?
  5. The conclusion should mention future plans with this study also ways to improve this study. Again, how this algorithm can mitigate the problem of false diagnosis of cases with milder symptoms of CKD.
  6. The authors need to bring in some graphs emphasizing the significant points/outcomes of this research. Those can be easily prepared using the data from Tables 2 and 3.
  7. Authors need to include a flow chart showing the whole procedure of the validation process.

Author Response

Point 1: In the abstract the authors claimed that: “adequate performance and can be used to obtain estimates of the prevalence of CKD”, a sensitivity of 51% can not be categorized as adequate. Authors need to rephrase their sentence about “adequate performance”.

Response 1: We modified the sentence in the abstract as follows: “adequate performance especially with regard to specificity and for more advanced CKD” (lines: 26-27)

Point 2: Authors claimed that “milder stages of CKD might remain undetected”, although the algorithm is cost-effective the authors need to explain what is the solution in case “milder CKD might remain undetected”.

Response 2: we added the following statement: “Depending on the setting and on whether a broader or stricter definition of CKD is desired, the algorithm for the prediction of more severe CKD could be used, resulting in an increase in both sensitivity and specificity” (lines: 186-189)

Point 3: In the discussion, the authors discussed several validation studies but only a few of them were from the last 5 years. Authors need to do a more extensive literature survey and include all significant similar validation studies which happened in the last five years. Need to mention the strength and shortcomings of those studies as well.

Response 3: we added the following statement: “Moreover, an Italian review (published in 2019) concludes on the need for further effort to improve algorithms for identification of CKD especially for early stages [11]. … In a study conducted in Canada (2020) in a population of just over 400 individuals, different algorithms were validated to identiy patients with eGFR<30 mL/min/1.73 m2, not on kidney replacement treatment (dialysis or kidney transplantation), authors found sensitivity and PPV higher and specificity and VPN lower than in our study [13].” (lines: 208-216)

Point 4: What are probable solutions for the limitations of this study which were mentioned by the authors?

Response 4: The limitations of our algorithm are intrinsic to the nature of the administrative data used; as such, it is difficult to envision a realistic solution for such limitations. With regard to the issue related to lower sensitivity, please see our response to point #2

Point 5: The conclusion should mention future plans with this study also ways to improve this study. Again, how this algorithm can mitigate the problem of false diagnosis of cases with milder symptoms of CKD.

Response 5: With regard to the issue related to lower sensitivity, please see our response to point #2. We have added a statement on possible future applications of our model to the revised manuscript: “Future studies will include the implementation of more detailed laboratory data such as serum cystatin C and urinary albumin excretion as well as the extension of our model to a multi-regional setting to obtain and compare estimates of CKD at the national level.” (lines: 249-252)

Point 6: The authors need to bring in some graphs emphasizing the significant points/outcomes of this research. Those can be easily prepared using the data from Tables 2 and 3.

Response 6: We presented the validation measures by age in figure 2 and eliminate those measures from the table 3

Point 7: Authors need to include a flow chart showing the whole procedure of the validation process.

Response 7: We added the Flow-chart of study population selection (figure 1)

Author Response

Major revisions:

Point 1: Further description of the laboratory methods used for measuring creatinine (i.e. by Jaffe vs enzymatic methods) are needed.

Response 1: We modified the methods accordingly: “Serum creatinine was measured with the enzymatic method” (lines 92-93)

Point 2: The methods section of this manuscript currently relies heavily on the authors’ previous paper detailing the algorithm. It is effectively unreadable for those not familiar with Italian health registry data. A clear description of the data used (similar to Table 1 in their 2020 BMC Nephrol article) should be included in either the main body of the manuscript or supplemental material.

Response 2: We added the table 1 of the article of 2020 in the supplementary tables

Point 2: A section in the methods section about the ethics (IRB approval, STROBE checklist adherence, etc.) of the study should be included.

Response 3: we added the IRB approval number to the Methods (lines 66-67)

Point  4: A flow diagram of participant inclusion (described in the first paragraph of the results) should be included.

Response 4: We added the Flow-chart of study population selection (figure 1)

Point 5: If the authors include statements about differences between groups (e.g., “men being on average older than women” lines 117-118, “the prevalence was higher among men compared with women” lines 123-124), appropriate hypothesis testing should be included to support these assertions.

Response 5: to the request of the Reviewer, we added statistical tests for the comparison between groups (lines: 125, 132)

Point 6: The authors should describe how the prevalence of CKD determined in their study (11.1%) compares to other estimates in the literature.

Response 6: We added a paragraph in the Discussion (lines: 167-178), as follows:

“Overall, we found a prevalence of CKD of about 11%. This figure is at odds with previous reports of CKD prevalence in the Italian population. For instance, the prevalence of CKD in the Northeastern Italy INCIPE study was about 13% overall, but only 6.7% when considering stages G3 or worse, which is the definition of CKD used in our study.[1] This discrepancy might in part be due to differences in the study populations, with the INCIPE cohort including a random sample of the general population whereas our sample was selected based on having at least one laboratory measurement performed at the Gemelli Hospital: this approach might have selected a less healthy population in our study. Another potential explanation is that a single determination of serum creatinine might have misclassified some INCIPE participants as non-CKD. Similar considerations apply to the CARHES study, a national survey reporting an even lower prevalence of CKD stage G3 or worse of about 2.9%.”

Point 7: The hypothesis that older participants have prevalence of CKD overestimated (lines 192- 193), while certainly plausible, is not supported by the finding of both decreased specificity and sensitivity.

Response 7: we would respectfully disagree with the Reviewer on this point: although it is true that specificity also decreases for higher age groups, such change is small in magnitude and within the confidence interval boundaries of the lower age groups, suggesting a fluctuation by chance rather than a systematic phenomenon; on the other hand, sensitivity drops substantially among older participants. For a comparison, specificity for GFR <60 mL/min/1.73 m2 changes from 98.1 (CI95% 97.8, 98.4) to 93.6 (CI95% 91.7, 95.4), whereas sensitivity goes from 86.0 (CI95% 57.7, 62.3) to 46.3 (CI95% 42.4, 50.2). Taken together, we maintain that these findings could be explained by overestimation of CKD among older participants.

we have however inserted a paragraph in the Discussion (lines: 228-232), as follows:

However, alternative explanations could be hypothesized; for instance, older individuals might already have ticket exemptions for other conditions, or not receive a discharge diagnosis of CKD due to the number and severity of other comorbidities (thus diminishing the likelihood of being flagged with those criteria)

Point 8: The limitations of this paper should address the inability to extrapolate to areas without the excellent accessible health data available in Lazio (e.g. in the USA).

Response 8: We added this point to the limitations list (lines: 243-244): “; furthermore, our model can only be applied in settings characterized by the availability of sufficiently detailed healthcare data”

Point 9: The authors should describe future directions for this research, potentially including a prospective validation cohort.

Response 9: We added a sentence on potential future directions (lines: 249-252): “Future studies will include the implementation of more detailed laboratory data such as serum cystatin C and urinary albumin excretion as well as the extension of our model to a multi-regional setting to obtain and compare estimates of CKD at the national level.”

Minor revisions:

Point 10: In the second paragraph of the introduction (lines 45-46), the phrase “laboratory/imaging criterion (renal damage or eGFR <60)” should be revised to read “biochemical or imaging evidence of impaired kidney structure or function.”

Response 10: we modified the definition accordingly: “biochemical or imaging evidence of impaired kidney structure or function lasting for at least 3 months” (lines: 46-49)

Point 11: A review of the staging of CKD would be helpful given that this is a general medicine journal whose audience may not be nephrologists. The KDIGO practice guideline table would be enough here.

Response 11: we added the KDIGO staging criteria in a Supplementary Table as request by the Reviewer and we modified the manuscript accordingly (lines: 49-52)

Point 12: The word “kidney” should be used rather than “renal” or “nephro-“ throughout, in keeping with KDIGO consensus guidelines (https://doi.org/10.1016/ j.kint.2020.02.010).

Response 12: we modified the wording throughout the manuscript according to the KDIGO Consensus Conference

Point 13: The authors should avoid the phrase “gold-standard” for evaluation of kidney function, which would require measured GFR (e.g. by iohexol or inulin clearance) rather than estimated GFR from creatinine. A descriptor such as “working standard” might be more accurate.

Response 13: as suggested by the Reviewer, we modified the manuscript accordingly

Point 14: There appears to be a type in Table 2 – I suspect the far right column should be headed “CKD [AL]” rather than “CKD [GS].”

Response 14: We correct the column name.

Point 15: Is the top half of Table 2 all participants with eGFR <60 or participants with eGFR 30- 60? I suspect the second.

Response 15: The table is correct. As explained in the paragraph “2.3. Measure of kidney function and definition of CKD.” the validation is on patient with eGFR <60 or eGFR <30. The algorithms proposed consider patients with CKD eGFR <60 (stage 3-5) or patients with advanced CKD eGFR <30 (stage 4-5). Those cut-off are selected according to the literature (bibliography 9-13)

Reviewer 3 Report

Dear Author,

I have few questions which could be helpful to improve the quality of your manuscript.

  1. The reference for the CKD and advanced CKD should be included (Line no 87-89)
  2. What is the proportion of underline causes of CKD such as diabetes, hypertension, obesity, etc.?
  3. What percent of participants without underlying conditions have CKD?
  4. Do you have any data about the CKD consequences such as anemia, dyslipidemia, secondary hypertension, and their prevalence according to the stage of CKD and any differences between sexes?

Author Response

Dear Reviewer

thank you for your comments.

Below you can find our answers point by point 

  • The reference for the CKD and advanced CKD should be included (Line no 87-89)
    • We added the reference as requested
  • What is the proportion of underline causes of CKD such as diabetes, hypertension, obesity, etc.?
    • Unfortunately, we do not have information on underlying causes of CKD or comorbidities, as our study is a validation of an already described algorithm for detection of CKD based on administrative data
  • What percent of participants without underlying conditions have CKD?
    • Please see our reply to previous point
  • Do you have any data about the CKD consequences such as anemia, dyslipidemia, secondary hypertension, and their prevalence according to the stage of CKD and any differences between sexes?
    • Please see our reply to previous point. We agree with the Reviewer that this information would be important, however validation studies of administrative algorithms usually do not supply this kind of information (12,13)

Round 2

Reviewer 1 Report

In response to points number 6 and 7, the authors have answered that they introduced figures 1 and 2, Those figures are not present or available in the revised manuscript or in supplementary files. 
Authors need to provide those figures before the manuscript gets accepted. The rest of the revision is acceptable. 

Author Response

Dear Reviewer,
thank you very much for helping us improve our manuscript.
As you requested, we have included the figures in the main file of the article.
Best Regards

Reviewer 2 Report

We appreciate the authors' thoughful responses and improvements. Figure 1 does not appear to have been included in the submission; once this figure is included the manuscript will be suitable for publication.

Author Response

Dear Reviewer,
thank you very much for helping us improve our manuscript.
As you requested, we have included the figure in the main file of the article.
Best Regards

Reviewer 3 Report

Thank you for putting the reference. I find this paper is well presented and could be interesting for nephrology. Therefore, I endorse this paper to publish.

Thank you!

BP